# Molecular Detection of *Anaplasma phagocytophilum* in Cats in Europe and Associated Risk Factors

**DOI:** 10.3390/ani14162368

**Published:** 2024-08-15

**Authors:** Vera Geisen, Nikola Pantchev, Yury Zablotski, Olga Kim, Majda Globokar Vrhovec, Katrin Hartmann, Michéle Bergmann

**Affiliations:** 1LMU Small Animal Clinic, Centre for Clinical Veterinary Medicine, LMU, D-80539 Munich, Germany; y.zablotski@med.vetmed.uni-muenchen.de (Y.Z.); hartmann@lmu.de (K.H.); n.bergmann@medizinische-kleintierklinik.de (M.B.); 2IDEXX Laboratories, D-70806 Kornwestheim, Germany; nikola-pantchev@idexx.com (N.P.); olga-kim@idexx.com (O.K.); majda-globokarvrhovec@idexx.com (M.G.V.)

**Keywords:** vector-borne diseases, feline, anaplasmosis, PCR, Sweden, Germany, Northern Europe

## Abstract

**Simple Summary:**

Although *Anaplasma* (*A.*) *phagocytophilum* infection in cats is considered to be less frequent compared with dogs, there is evidence indicating that the risk of the infection in cats might be underestimated. The study aimed to find out if infections in cats are underestimated and to discover which factors increase the risk of infection. Blood samples of 1015 cats across Europe were tested for *A. phagocytophilum* DNA from 2017 to 2022. The number of samples sent for testing increased over the 6 years. *Anaplasma phagocytophilum* DNA was found in 76 out of 1015 cats (7.5%). Infections were more common in Northern Europe than in Central or Southern Europe. During summer, the number of positive samples was significantly higher compared with winter (*p* = 0.047). The risk for *A. phagocytophilum* infection in cats should not be underestimated, especially in Northern Europe. Preventing tick bites is essential for cats’ health all over Europe, not just in the Mediterranean regions.

**Abstract:**

Infections with *Anaplasma* (*A.*) *phagocytophilum* in cats seem to be rare. The study aimed to determine whether infections in cats are underestimated and to identify the risk factors for infection. Blood samples of 1015 cats across Europe (2017–2022), sent to IDEXX Laboratories, Germany, were tested for *A. phagocytophilum* DNA. The influence of the cats’ origin on *A. phagocytophilum* infection was assessed by univariable analysis, while multivariable logistic regression evaluated associations with the cats’ sex and age, and the years, and seasonality of the samples’ submission. Furthermore, univariable linear regression was used to determine patterns in PCR orders. The number of submitted samples increased significantly during the 6 years (*p* = 0.042). *Anaplasma phagocytophilum* DNA was detected in 76/1015 of cats (7.5%, 95% CI 6.0–9.3%). Infections were significantly more common in Northern compared to Central (*p* < 0.001, OR: 8.70) and Southern Europe (*p* < 0.001, OR: 39.94). A significantly higher likelihood for infections during the summer compared with winter (*p* = 0.047, OR: 3.13) was found. Bacteremia with *A. phagocytophilum* in European cats is not uncommon. *Anaplasma phagocytophilum* infection should be considered an important risk, particularly in Northern Europe. Effective tick prevention is crucial for managing feline health across Europe, not just in the Mediterranean region.

## 1. Introduction

*Anaplasma* (*A.*) *phagocytophilum* is an emerging pathogen that affects humans and various other species, including dogs and cats [1,2,3,4]. *Anaplasma phagocytophilum* is a Gram-negative, obligate intracellular bacterium [5]. Its main vector in Europe is *Ixodes* (*I.*) *ricinus* [6]; however, other *Ixodes* ticks (e.g., *I. trianguliceps, I. hexagonus*, and *I. ventalloi*) also have been documented as vectors [7,8,9,10]. The exact role of *A. phagocytophilum* as a pathogen in cats is not fully clear. Some cats are asymptomatically infected, and a few cats develop clinical signs, which are primarily unspecific, including lethargy, fever, and anorexia [11,12,13,14]. Main laboratory changes in cats (if present) are thrombocytopenia, leukopenia, leukocytosis, lymphopenia, and anemia [11,12,15]. Since anaplasmosis is an acute disease with a short incubation period, accurate diagnosis relies on direct detection of the pathogen (via PCR or the identification of morulae in neutrophilic granulocytes) in combination with compatible clinical signs and/or laboratory changes [16].

Although *A. phagocytophilum* infection in cats is considered to be less frequent compared with dogs [16,17,18], there is evidence indicating that infection risk in cats might be underestimated [12,13,15,19].

As observed in dogs, the prevalence of antibodies against *A. phagocytophilum* in cats is notably high. The antibodies’ prevalence in cats was determined to be 4.5–33.3% in Italy [20,21], 30% in Austria [12,22], 24% in Switzerland [12], 9.1–23% in Germany [12,23,24], 22.1% in Sweden [25], and 1.8–8.4% in Spain [26,27,28,29]. However, considering that anaplasmosis is an acute, self-limiting disease, the presence of antibodies is not indicative of a clinical manifestation compared with direct detection of the pathogen [30,31].

Direct detection of the pathogen via PCR yielded (less often) positive results. In cats in Italy, prevalence ranged from 0–23.1% [20,21,32]; in Portugal, from 0–0.6% [33,34]; in Germany, from 0.3–3% [12,17,23,24]; in the United Kingdom (UK), 1.7% [35], and in Spain, from 0–1% [26,29,36]. However, the majority of cited studies are not up to date, and a comparative study of *A. phagocytophilum* infections in cats across European countries has not been conducted.

In the meantime, several clinical case reports and series on clinical manifestations of feline anaplasmosis have been published, including a recent study involving 27 *A. phagocytophilum*-positive cats from Germany (molecular prevalence: 3%), Switzerland (molecular prevalence: 10%), and Austria (molecular prevalence: 8%) [12]. The aims of the present study were to investigate whether the risk of *A. phagocytophilum* bacteremia in cats in Europe is underestimated and to compare the results across Europe. Furthermore, the influence of cats’ gender and age, and the years of the samples’ submission and seasonality (spring, summer, autumn, winter) on *A. phagocytophilum* infection was evaluated. Additionally, it was investigated wether a pattern in PCR submissions could be observed over time.

## 2. Materials and Methods

Blood samples of 1015 cats were tested for *A. phagocytophilum* DNA. Blood samples derived from submissions by veterinarians from different European countries within 6 years (2017–2022) to a commercial laboratory (IDEXX Laboratories, Kornwestheim, Germany) (Table 1). A single *A. phagocytophilum* PCR was requested for 651 samples, while 26 samples were submitted for a tick panel, 242 samples for an anemia panel (available since 2022), and 96 samples for a fever panel (available since 2022), all including a PCR for *A. phagocytophilum*. Further information about the reasons for submission was not available due to the nature of the study (a retrospective evaluation of laboratory submissions), but it can be assumed that a clinical suspicion was raised by the submitting veterinarian in the majority of the cases. All samples were included in which information on the date of sample’s submission, country, postal code, and city of the submitting veterinarian, age and sex of the sampled cats; and the result of the *A. phagocytophilum* PCR could be obtained from the laboratory’s computer system. Samples from outside Europe were excluded.

*A. phagocytophilum* DNA was amplified by real-time PCR from whole blood according to Dyachenko and colleagues (2012) at IDEXX Laboratories Inc. Kornwestheim, Germany [37]. The extraction of total DNA from whole blood was carried out utilizing the QIAamp DNA Blood BioRobot MDx kit (QIAGEN, Hilden, Germany), adhering to the guidelines provided by the manufacturer. Real-time PCR was carried out utilizing the LightCycler 480 system (Roche, Mannheim, Germany), using custom-designed forward and reverse primers in addition to hydrolysis probes. The genes selected as the target genes for detection of the pathogen via real-time PCR was *msp2* of *A. phagocytophilum* (accession number DQ519570). The real-time PCR assays were shown to have a reproducible average analytical sensitivity of 10 DNA molecules per reaction.

Statistical analysis was performed using R Version 4.3.1. Robust linear regression was used to determine whether there was a trend of either an increase or decrease in PCR requests submitted by veterinarians over time. To determine associations between the cats’ origin (country and geographic distribution) and *A. phagocytophilum* infection, univariable logistic regression was performed. For these analyses, Europe was divided into three geographical regions: Northern Europe (Sweden, Finland, Denmark, and Norway), Central Europe (Netherlands, UK, Germany, Poland, Switzerland, Austria, Hungary, Belgium, Czech Republic, and Slovakia), and Southern Europe (France, Spain, Italy, and Slovenia). Associations of the cats’ sex and age, the years of sample’ submission (years 2017–2019 versus 2020–2022), and seasonality with *A. phagocytophilum* infection were evaluated by multivariable logistic regression. Since all four predictors were considered clinically relevant, no backward selection of the variables was performed. To exclude multicollinearity among the predictors, the generalized variance-inflation factors (GVIFs) were calculated for the logistic regression. Due to the lower number of samples in some years, data were consolidated into 2 periods of equal length: period 1: 2017–2019 and period 2: 2020–2022. In countries with more than 100 samples submitted overall, an association between the period of the samples’ submission (2017–2019 versus 2020–2022) and *A. phagocytophilum* infection was evaluated using bivariable logistic regression. For all analyses, a *p*-value of <0.05 was considered significant.

## 3. Results

### 3.1. Population

Of the 1015 sampled cats, 302/864 (35.0%) were female and 562/864 (65.0%) were male; sex was unknown in 151 cats. The age was known in 781 cats and ranged from 0.17–23 years (median: 5 years); the age of 234 cats was unknown. Of the 1015 cats, 411 originated from Germany (40.5%), 132 from Sweden (13.0%), 89 from Switzerland (8.8%), 79 from the UK (7.8%), 57 from Italy (5.6%), 49 from the Netherlands (4.8%), 46 from Austria (4.5%), 41 from Spain (4.0%), 30 from Finland (3.0%), 21 from Denmark (2.1%), 19 from France (1.9%), 12 from Belgium, 12 from Norway (1.2%), 8 from Poland (0.8%), 4 from Hungary (0.4%), 3 from Czech Republic (0.3%), and 1 each from Slovenia and Slovakia (0.1%) (Figure 1).

### 3.2. Submissions to the Laboratory

PCR orders submitted to the laboratory increased significantly over the past 6 years (*p* = 0.002) (Figure 2) with a mean increase of 34 tests per year. Since the numbers exhibited an outlier-like explosion in 2022, robust linear regression was used to determine the pattern of submission, in which the outliers had less weight.

### 3.3. Anaplasma Phagocytophilum-Infected Cats

*Anaplasma phagocytophilum* DNA was detected in 76/1015 of cats (7.5%, 95% CI: 6.0–9.3%). Of 302 female cats, 21 (6.9%) were infected, and of 562 male cats, 48 (8.5%) were infected. There was no significant difference in sex between *A. phagocytophilum*-negative and -positive cats (multivariable logistic regression: *p* = 0.600) (Table 1).

The median age of *A. phagocytophilum*-infected cats was 3 years (range: 0.25–16 years); that of *A. phagocytophilum*-noninfected cats was 5 years (range: 0.17–23 years). The risk of *A. phagocytophilum* infection significantly decreased with the cats’ age (*p* = 0.013, odds ratio (OR): 0.92) (Table 1, Figure 3).

The highest percentage of *A. phagocytophilum* infections was detected in Sweden (38/132; 28.8%) and Finland (6/30; 20%). This was only outnumbered by Hungary, where, however, only 4 samples were tested (2 out of 4 were positive) (Figure 1). Figure 4 shows the proportion of *A. phagocytophilum*-infected cats out of the number of tested cats, and the percentage for each year.

*Anaplasma phagocytophilum* infection was significantly more common in Northern Europe when compared with Central (*p* < 0.001, OR: 8.70) and Southern Europe (*p* < 0.001, OR: 39.94) (Figure 1). While throughout Europe, the percentage of *A. phagocytophilum* infections significantly decreased over the past 3 years (2020–2022) compared with the previous 3 years (2017–2019) (*p* = 0.017, OR: 2.09) (Figure 4), it did not change significantly in Germany (*p* = 0.471) and Sweden (*p* = 0.824) (Figure 5).

Multivariable logistic regression revealed a significantly higher likelihood for *A. phagocytophilum* infection during summer compared with winter (*p* = 0.047, OR: 3.13) (Figure 6, Table 1). Other significant seasonal differences were not observed.

## 4. Discussion

PCR requests for *A. phagocytophilum* submitted to the laboratory significantly increased over the past 6 years, while the percentage of infection significantly decreased. This trend is likely attributed to the growing awareness among veterinarians regarding *A. phagocytophilum* infection and its integration in laboratory panels.

In total, *A. phagocytophilum* DNA was detected in 76 out of 1015 (7.5%) cats. The notably high percentage of *A. phagocytophilum*-positive PCR results was unexpected. The percentage was high, particularly in Northern Europe (25.1% (8.3–28.8%)), which can be explained by the northward expansion of the vectors. Several decades ago, *Ixodes* ticks were scarce in Northern Europe [38,39], but their range has expanded northward over time. This was accompanied by a significant increase in the population density of ticks, as seen, for example, in Sweden [40,41,42], the country with the highest number of *A. phagocytophilum* infections in the present study. Steadily rising temperatures are (at least partially) associated with the northward migration of the ticks’ hosts, such as deer and moose [40]. Consequently, this migration facilitates the broader distribution of *Ixodes* ticks, which require temperatures of more than 5–7 °C for their activity [43,44]. On the other hand, surprisingly, a recent study discovered that *I. ricinus* in Sweden exhibited activity even at temperatures as low as −5 °C [45]. Furthermore, one study revealed that *A. phagocytophilum* can induce the expression of an antifreeze glycoprotein gene in *Ixodes* ticks, enhancing the ticks’ cold resistance [46]. A further explanation for the high prevalence of *Ixodes* ticks in Northern Europe is the explosive proliferation of the ticks’ main host, the red deer. In the early 1990s, red deer experienced a rapid increase in population due to an outbreak of scabies in red foxes, leading to a reduction in the deer’s natural predators [47]. Another important fact is the high prevalence of *A. phagocytophilum* in both the vector and reservoir hosts. The prevalence of *A. phagocytophilum* in *I. ricinus* by PCR was reported to be 0.7–15.0% in Sweden [48,49], 3.5–9.6% in Finland [50,51], and 40.5% in Denmark [52], in comparison with only 3.1–6.5% in Germany [53,54,55,56]. A molecular prevalence of 26.3% was observed in Swedish moose [25], 23.9% in Swedish cattle [57], and even 42.6% in Danish roe deer [58]. A further point to consider regarding the high number of *A. phagocytophilum-*positive cats in Northern Europe is their lifestyle. Many cats in this region have outdoor access, making them more susceptible to tick infestations and thus infection [59]. Additionally, many studies have shown that *I. ricinus* prefers wooded areas, which are very common in Northern European countries and thus predispose this area to a high(er) tick prevalence [60,61,62]. In this context, the afforestation of open vegetation types, as are currently occurring through some climate protection projects in Scandinavia, have likely also led to increased tick density [62].

In Central Europe, the rate of *A. phagocytophilum* infections was relatively high (3.9%, 16/411) paralleling the results of a recent study that documented a comparable rate of 3% (18/619) [12]. This contrasts starkly with earlier research, where the molecular prevalence ranged from only 0.3–0.4% [12,17,23,24]. Also, the prevalence of *A. phagocytophilum* in *Ixodes* ticks in Germany exhibited a notable increase, escalating from 1% in 2004 [63] to as high as 6.5% in studies conducted between 2010 and 2021 [53,54,55,56]. However, tests have improved over the years, which has potentially impacted the increase in positive results as well.

In the present study, a notably low percentage of infected cats was observed in Southern Europe, with only 1 out of 118 positive samples (0.8%). This finding contrasts with an older study conducted in Northern Italy, which reported a notably higher prevalence of 23.1% [20]. However, all cats sampled in that study in 2014 were stray cats, which were certainly not protected against ectoparasites. Owned cats, in comparison, are more likely to receive regular ectoparasite prophylaxis, especially in regions endemic for chronic persistent vector-borne diseases in Southern Europe; this would also explain the controversially low number of *A. phagocytophilum* infections in the current study and in multiple previous studies (≤1%) [21,26,28,32,33,34,36,64]. Another contributing factor for the low rate in Southern Europe is the lower prevalence of *Ixodes* ticks in Mediterranean regions, since this ticks require higher humidity levels to thrive [43].

So far, relatively little information is available about the association of the age of cats with *A. phagocytophilum* bacteremia. In the present study, *A. phagocytophilum*-infected cats were relatively young (median age: 3 years), and there was a significant decrease in positive rates with increasing age (Figure 2). The median age in dogs is 7–8 years [2,65,66]. The low median age in cats could indicate an age-related resistance in older cats, similar to what is observed with *Bartonella* spp. infection in cats [67]. This could potentially be attributed to a differential function of the cat’s immune system as opposed to that of dogs [68].

*Anaplasma phagocytophilum* infection was significantly more likely in summer compared with winter. This observation aligns with the reported tick activity [69] and is similar to findings of other studies examining at the seasonal presence of *A. phagocytophilum* [66,70]. However, it is noteworthy that some studies reported a bimodal distribution of *A. phagocytophilum* infection, with peaks occurring in spring and autumn [65].

The limitation of the study is the preselection of samples, as only those submitted to the laboratory were examined, thereby deviating from a purely prevalence-based investigation.

## 5. Conclusions

In conclusion, the present study highlights that infections with *A. phagocytophilum* confirmed by PCR in cats are more common than expected. *Anaplasma phagocytophilum* infection should be considered as important infections, particularly in Northern Europe. Therefore, implementing effective tick prevention measures is crucial for the management of feline health and are as important in non-Mediterranean regions as in Mediterranean ones.

## Figures and Tables

**Figure 1 animals-14-02368-f001:**
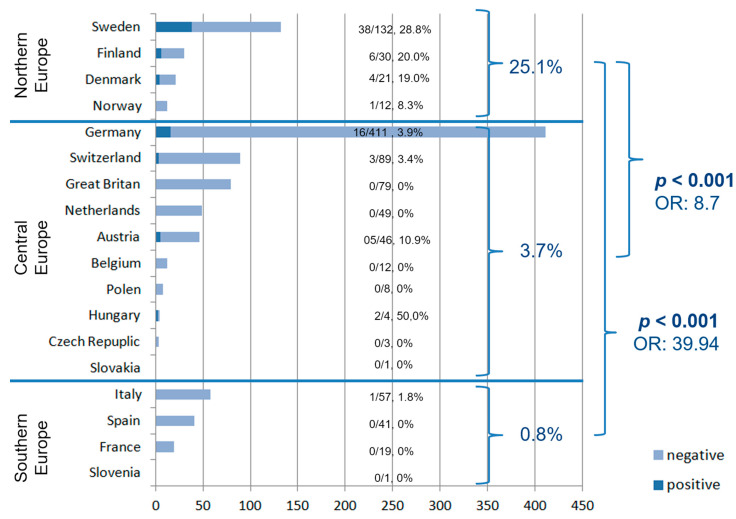
Number of submitted samples and number and percentage of *Anaplasma* (*A.*) *phagocytophilum*-infected in different European countries. Univariable logistic regression revealed that the likelihood of *A. phagocytophilum* infection was significantly higher in cats from Northern Europe in comparison with cats from Central (*p* < 0.001, OR: 8.70) and Southern Europe (*p* < 0.001, OR: 39.94). Bold values indicate significance. A *p*-value of <0.05 was considered significant. OR, odds ratio; *p*, *p*-value.

**Figure 2 animals-14-02368-f002:**
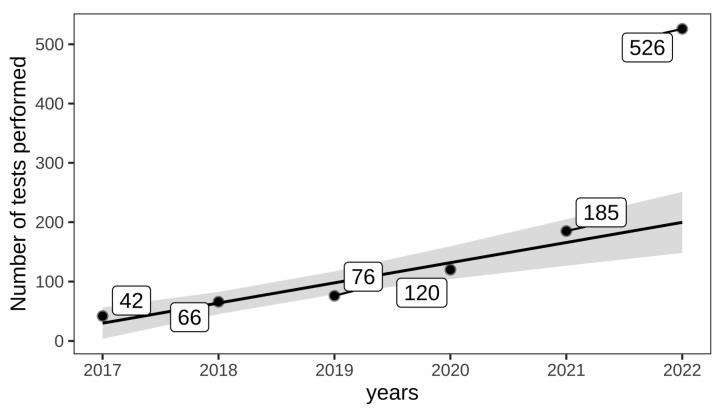
Number of *Anaplasma* (*A.*) *phagocytophilum* PCR requests submitted by veterinarians to the laboratory over time (2017–2022). Robust linear regression revealed a significant increase over the years (*p* = 0.002), with a mean increase of 34 tests performed per year. In 2022, the numbers exhibited an outlier-like explosion; however, since robust linear regression was used to determine the pattern of submission, the outliers had less weight.

**Figure 3 animals-14-02368-f003:**
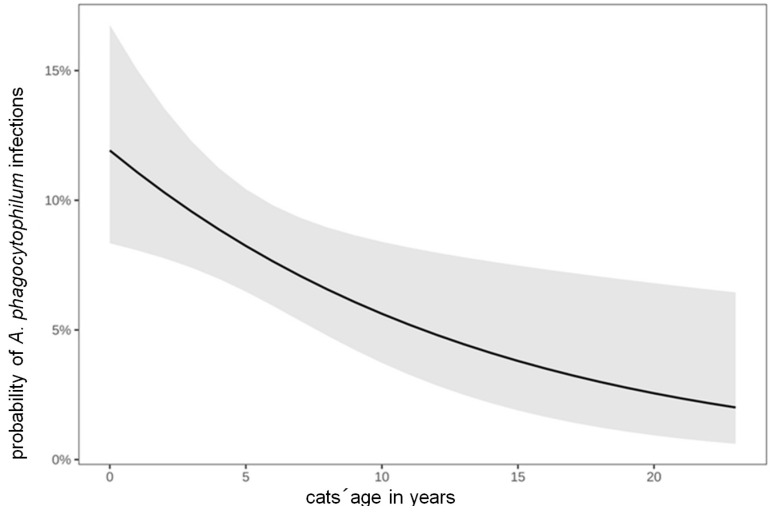
Association between cats’ age in years and the number of *Anaplasma* (*A.*) *phagocytophilum* infections. Multivariable logistic regression revealed that the risk for *A*. *phagocytophilum* infection significantly decreased with the cats’ age (*p* = 0.013, OR: 0.92). A *p*-value of <0.05 was considered significant. OR, odds ratio; *p*, *p*-value.

**Figure 4 animals-14-02368-f004:**
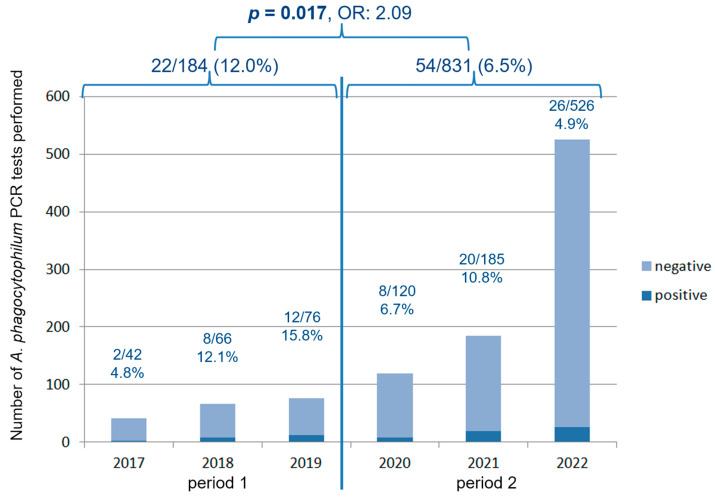
*Anaplasma* (*A.*) *phagocytophilum* bacteremic cats per number of tested cats and percentages. Despite the fact that the number of tests increased over the years, the percentage of *A*. *phagocytophilum*-positive cats decreased. Multivariable logistic regression revealed a significantly higher odds ratio for being infected in the early years (2017–2019) compared with later years (2020–2022). A *p*-value of <0.05 was considered significant. OR, odds ratio; *p*, *p*-value.

**Figure 5 animals-14-02368-f005:**
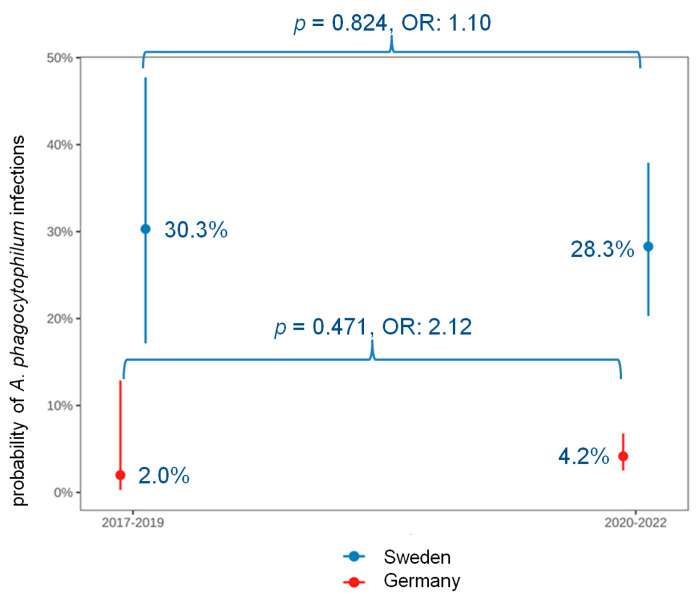
Association between years of the samples’ submission (2017–2019 versus 2020–2022) and *Anaplasma* (*A.*) *phagocytophilum* infection in Germany and Sweden (countries in which the highest numbers of samples were tested). The figure shows the predictive probability of *A. phagocytophilum* infections and the 95% confidence intervals of each predictor. Bivariable logistic regression revealed no significant changes within the past 3 years (2020–2022) compared with the previous 3 years (2017–2019). A *p*-value of <0.05 was considered significant. OR, odds ratio; *p*, *p*-value.

**Figure 6 animals-14-02368-f006:**
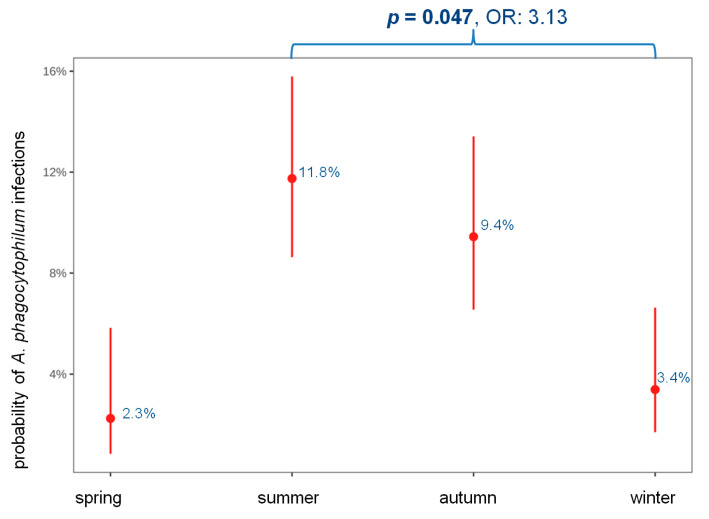
Association between seasonality and *Anaplasma* (*A.*) *phagocytophilum* infection. The figure shows the predictive probability of *A. phagocytophilum* infections and the 95% confidence intervals of each predictor. Multivariable logistic regression revealed a significantly higher likelihood for *A. phagocytophilum* infections in summer than in winter (*p* = 0.047, OR: 3.13). There were no significant differences among the other seasons. Bold values indicate significance. A *p*-value of <0.05 was considered significant. OR, odds ratio; *p*, *p*-value.

**Table 1 animals-14-02368-t001:** Association among cats’ age and sex, the period of the samples’ submission (2017–2019 versus 2020–2022), seasonality, origin, and number of *Anaplasma* (*A.*) *phagocytophilum* PCR referrals submitted by veterinarians to the laboratory over time.

Risk Factor	*A. phagocytophilum-*Infected/Number of Tested Cats (%)	Constrasts	Multivariable Analyses		
			OR	95% CI	*p*-Value
Age (n = 781)		Years	0.92	0.86–0.98	**0.019**
Sex (n = 864)	M: 48/562 (8.5)	F/M	0.85	0.47–1.53	0.600
	F: 21/302 (6.9)				
Period of submission of the sample (n = 1015)	2017–2019: 22/184 (12.0)	(2017–2019)/(2020–2022)	2.09	0.26–0.88	**0.017**
	2020–2022: 54/831 (6.5)				
Seasonality		Summer/spring	4.10	1.00–16.9	0.051
	spring: 4/178 (2.3)	Autumn/summer	0.80	0.36–1.78	0.900
	summer: 37/315 (11.8)	Winter/summer	0.32	0.10–0.99	**0.047**
	autumn: 27/286 (9.4)	Autumn/spring	3.30	0.78–13.9	0.140
	winter: 8/236 (3.4)	Winter/autumn	0.40	0.12–1.27	0.200
		Winter/spring	1.31	0.25–6.78	>0.900
			**Univariable analysis**		
Origin	NE: 49/195 (25.1)	NE/CE	8.70	4.74–16.00	**<0.001**
	CE: 26/702 (3.70)	NE/SE	39.94	3.68–433.70	**<0.001**
	SE: 1/118 (0.85)	CE/SE	4.59	0.42–50.60	0.296

Bold values indicate significance. A *p*-value of < 0.05 was considered significant. CE, Central Europe (Netherlands, United Kingdom, Germany, Poland, Switzerland, Austria, Hungary, Belgium, and Czech Republic); CI, confidence interval; F, female; M, male; NE, Northern Europe (Sweden, Finland, Denmark, and Norway); n, number; OR, odds ratio; SE, Southern Europe (France, Spain, Italy, and Slovenia).

## Data Availability

The data presented in this study are available upon request from the corresponding author.

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
