# Peer review of "Molecular Detection of Anaplasma phagocytophilum in Cats in Europe and Associated Risk Factors"

_animals, 2024, doi:10.3390/ani14162368_

Round 1

Reviewer 1 Report

Comments and Suggestions for Authors

Well-written manuscript with important regional information about Anaplasma phagocytophilum in cats from European countries.

No major issues were detected, only a minor detail was repeated throughout the manuscript. The pathogen name cannot be abbreviated when a sentence starts. 

After correcting the minor issue mentioned above, the manuscript can be suited for publication.

Author Response

Dear Reviewers,

we are grateful for the time and effort you put into reviewing the paper “Molecular Detection of Anaplasma phagocytophilum in cats in Europe and risk factors”. Your insightful comments and constructive feedback have been extremely helpful for improvement of the manuscript. We believe to have thoroughly addressed all points. Detailed remarks to the reviewers´ comments are below.

Thank you very much.

Reviewer 1

Comments and suggestions for authors:

  • Well-written manuscript with important regional information about Anaplasma phagocytophilum in cats from European countries.

We thank the reviewer for the positive comment.

  • No major issues were detected, only a minor detail was repeated throughout the manuscript. The pathogen name cannot be abbreviated when a sentence starts. 

Thank you, for pointing this out.

A.” was replaced by “Anaplasma” at all sentence starts throughout the manuscript.

Line  14: “Anaplasma

Line  27: “Anaplasma

Line  32: “Anaplasma

Line  39: “Anaplasma

Line  164: “Anaplasma

Line  165: “Anaplasma

Line  194: “Anaplasma

Line  282: “Anaplasma

Line  294: “Anaplasma

  • After correcting the minor issue mentioned above, the manuscript can be suited for publication.

We thank the reviewer for recommendation of the manuscript for publication.

Reviewer 2 Report

Comments and Suggestions for Authors

Work with cats is rare and very welcome.

The study aimed to determine whether infections in cats are underestimated and to identify risk factors for infection.

The introduction should be expanded to talk mainly about the body.

The materials and methods are well described.

The discussion should be expanded. Please comment more on the organisms and differences between localities, especially in terms of farms and environments.

In the discussion: In addition, further studies on the clinical role of A. phagocytophilum as pathogen in cats are necessary. That's a cliché and should be removed. All work always has to be continued.

The conclusion: Further studies on the clinical role of A. phagocytophilum as pathogen in cats are necessary. That's a cliché and should be removed. All work always has to be continued.

Author Response

Dear Reviewers,

we are grateful for the time and effort you put into reviewing the paper “Molecular Detection of Anaplasma phagocytophilum in cats in Europe and risk factors”. Your insightful comments and constructive feedback have been extremely helpful for improvement of the manuscript. We believe to have thoroughly addressed all points. Detailed remarks to the reviewers´ comments are below.

Thank you very much.

Comments and suggestions for authors:

  • Work with cats is rare and very welcome.

We fully agree with the reviewer´s comment.

  • The study aimed to determine whether infections in cats are underestimated and to identify risk factors for infection.

  • The introduction should be expanded to talk mainly about the body.

Thanks for pointing this out. The introduction was expanded.

Line 41-48: “The exact role of A. phagocytophilum as pathogen in cats is not fully clear. Some cats are asymptomatically infected and few cats develop clinical signs which are primarily unspecific, including lethargy, fever, and anorexia [11-14]. Main laboratory changes in cats (if present) are thrombocytopenia, leukopenia or leukocytosis, lymphopenia, and anemia [11,12,15]. Since anaplasmosis is an acute disease with short incubation period, accurate diagnosis relies on direct pathogen detection (via PCR or the identification of morulae in neutrophilic granulocytes) in combination with compatible clinical signs and/or laboratory changes [16].”

  • The materials and methods are well described.

We thank the reviewer for the positive comment.

  • The discussion should be expanded. Please comment more on the organisms and differences between localities, especially in terms of farms and environments.

Thanks for pointing this out. The discussion was expanded.

Line 249-255:  “Many cats in this region have outdoor access, making them more susceptible to tick infestations and thus infection [59]. Additionally, many studies have shown that I. ricinus prefers wooded areas, which are very common in Northern European countries and thus, predispose for a high(er) tick prevalence [60-62]. In this context, the afforestation of open vegetation types, as currently occurring through some climate protection projects in Scandinavia, likely also leads to increased tick density [62].”

  • In the discussion: In addition, further studies on the clinical role of A. phagocytophilum as pathogen in cats are necessary. That's a cliché and should be removed. All work always has to be continued.

The sentence was deleted.

  • The conclusion: Further studies on the clinical role of A. phagocytophilum as pathogen in cats are necessary. That's a cliché and should be removed. All work always has to be continued.

The sentence was deleted.

Reviewer 3 Report

Comments and Suggestions for Authors

The authors performed this study to detect Anaplasma phagocytophilum and risk factors related to this infection in cats. They did a good job. However, I need to be clarified for the following comments before further steps:

Line 2-3: I would suggest adding risk factors in the title.

Line 9: Full name of A. phagocytophilum?

Line 16: “p” from “p-value” should be in italics. Please correct this throughout the manuscript.

Line 66: The authors could arrange the methodology section properly.

Line 80-95: Please move this information to the results section.

Line 103: The “msp2” gene should be in italics.

Line 106-121: I suggest the authors mention the logistic regression in detail. For example, how did they select variables to create the regression model? Did they perform a collinearity test or not? If yes, how? If not, why? They should mention it for the logistic regression model.

Line 117-118: I was just wondering how and why the authors selected a three-year gap for the two sampling periods.

Author Response

Dear Reviewers,

we are grateful for the time and effort you put into reviewing the paper “Molecular Detection of Anaplasma phagocytophilum in cats in Europe and risk factors”. Your insightful comments and constructive feedback have been extremely helpful for improvement of the manuscript. We believe to have thoroughly addressed all points. Detailed remarks to the reviewers´ comments are below.

Thank you very much.

Reviewer 3:

Comments and suggestions for authors:

  • The authors performed this study to detect Anaplasma phagocytophilum and risk factors related to this infection in cats. They did a good job. However, I need to be clarified for the following comments before further steps:
  • Line 2-3: I would suggest adding risk factors in the title.

Thank you for pointing this out. We included “risk factors” in the title.

Line 2-3: “Molecular Detection of Anaplasma phagocytophilum in cats in Europe and Associated Risk Factors

  • Line 9: Full name of phagocytophilum?

Thank you for pointing this out. The name was spelled out.

Line 9: “Anaplasma

  • Line 16: “p” from “p-value” should be in italics. Please correct this throughout the manuscript.

Thank you for pointing this out. We changed “p” to “p” (italics letter) throughout the manuscript.

Line 16: “p

Line 28: “p

  • Line 66: The authors could arrange the methodology section properly.

The methodology section has been rearranged as suggested. The statistics part has been revised and the section about the sampled cats is now in the results section.

Line 125-128: “Since all four predictors were considered clinically relevant, no backwords-selection of variables was performed. To exclude multicollinearity among predictors, the generalized variance-inflation factors (GVIFs) were calculated for logistic regression.”

Line 137-144: Of the 1.015 sampled cats, 302/864 (35.0%) were female and 562/864 (65.0%) male; sex was unknown in 151 cats. Age was known in 781 cats and ranged from 0.17 to 23 years (median 5 years); age of 234 cats was unknown. Of the 1,015 cats, 411 originated Germany (40.5%), 132 from Sweden (13.0%), 89 from Switzerland (8.8%), 79 from UK (7.8%), 57 from Italy (5.6%), 49 from Netherlands (4.8%), 46 from Austria (4.5%), 41 from Spain (4.0%), 30 from Finland (3.0%), 21 from Denmark (2.1%), 19 from France (1.9%), 12 from Belgium and 12 from Norway (1.2%), 8 from Poland (0.8%), 4 from Hungary (0.4%), 3 from Czech Republic (0.3%), and 1 each from Slovenia and Slovakia (0.1%) (figure 1).

  • Line 80-95: Please move this information to the results section.

We moved this section including the first figure to the results section.

Line 137-144: Of the 1.015 sampled cats, 302/864 (35.0%) were female and 562/864 (65.0%) male; sex was unknown in 151 cats. Age was known in 781 cats and ranged from 0.17 to 23 years (median 5 years); age of 234 cats was unknown. Of the 1,015 cats, 411 originated Germany (40.5%), 132 from Sweden (13.0%), 89 from Switzerland (8.8%), 79 from UK (7.8%), 57 from Italy (5.6%), 49 from Netherlands (4.8%), 46 from Austria (4.5%), 41 from Spain (4.0%), 30 from Finland (3.0%), 21 from Denmark (2.1%), 19 from France (1.9%), 12 from Belgium and 12 from Norway (1.2%), 8 from Poland (0.8%), 4 from Hungary (0.4%), 3 from Czech Republic (0.3%), and 1 each from Slovenia and Slovakia (0.1%) (figure 1).

  • Line 103: The “msp2” gene should be in italics.

We thank the reviewer for making us aware of this mistake. “msp2” was changed to “msp2” (italics letter).

Line 113: “msp2

  • Line 106-121: I suggest the authors mention the logistic regression in detail. For example, how did they select variables to create the regression model? Did they perform a collinearity test or not? If yes, how? If not, why? They should mention it for the logistic regression model.

We thank the reviewer for this important point. The additional information was added.

Line 125-128: “Since all four predictors were considered clinically relevant, no backwords-selection of variables was performed. To exclude multicollinearity among predictors, the generalized variance-inflation factors (GVIFs) were calculated for logistic regression.”

  • Line 117-118: I was just wondering how and why the authors selected a three-year gap for the two sampling periods.

The number of samples was not always high enough to perform comparisons between individual years (e.g., only 42 submissions with 2 positive results in 2017). Therefore, we decided to consolidate the data into two periods of equal length.

Line 128-129: “Due to a lower number of samples in some years, data were consolidated into two periods of equal length.“

Round 2

Reviewer 1 Report

Comments and Suggestions for Authors

The manuscript is ready for publication.

Reviewer 2 Report

Comments and Suggestions for Authors

All the requested changes have been made. The article can be published.

Reviewer 3 Report

Comments and Suggestions for Authors

The authors addressed my comments. Thanks